# Impact of Nano Additives in Heat Exchangers with Twisted Tapes and Rings to Increase Efficiency: A Review

Younus Hamoudi Assaf [1], Abdulrazzak Akroot [1,*], Hasanain A. Abdul Wahhab [2,*], Wadah Talal [1], Mothana Bdaiwi [1] and Mohammed Y. Nawaf [1]

[1] Department of Mechanical Engineering, Faculty of Engineering, Karabük University, 78050 Karabük, Turkey

[2] Training and Workshop Center, University of Technology-Iraq, Ministry of Higher Education & Scientific Research, Baghdad 10066, Iraq

* Correspondence: abdulrazzakakroot@karabuk.edu.tr (A.A.); 20085@uotechnology.edu.iq (H.A.A.W.); Tel.: +964-781-130-9446 (H.A.A.W.)

**Abstract:** The heat exchanger is crucial to all systems and applications that use it. Researchers are primarily focused on improving this component's thermal conductivity to improve its efficiency. This was achieved by using one or more of the following strategies: inserting tapes with various shapes and numbers, inserting rings of various shapes and spacing between each, and transforming a basic liquid into a nanoliquid by adding nanomaterials with high conductivity and ultra-small particle sizes. Different types of nanomaterials were added in varying concentrations. In earlier studies, it was found that every increase in heat transfer was accompanied by a pressure drop at both ends of the exchanger. The amount of heat transferred and the pressure drop are affected by many factors, such as the torsion tape ratio, the pitch of the ring, and whether the pitch faces the direction of flow or not. Heat transfer rates can also be impacted by factors such as the length and angle of the wings, how many rings and tapes there are, and whether the rings and tapes contain holes or wings. In addition, the Reynolds number, the type, conductivity, and size of nanomaterials, and the base fluid used in the nanofluid affect this. It is possible for the shape of the exchanger tube, as well as varying rates of rise, to introduce such impacts. In this study, the factors, costs, and benefits of using any technology to increase the efficiency of the heat exchanger are reviewed so that the user can make an informed decision about the technology to use.

**Keywords:** heat exchanger; twisted tape; insert rings; Nusselt number; nanofluid

## 1. Introduction

Heat exchangers come in a variety of types and are often utilized in engineering and industrial purposes (such as in factories and power plants, heating and cooling equipment, and other industries, such as the electricity, chemical, energy-saving, and petroleum industries) [1,2]. According to the area, shape, and classification based on the function of the process and liquid-to-vapor phase-change exchanger classification, chemical evaporators can be classified into (i) the type of construction and (ii) how energy is supplied [3,4]. The classification of rebuilders and the tasks necessary for the exchange can be categorized in several ways to fit with the tasks carried out by the exchanger in various projects and systems [5,6]. There are no interactions in heat exchangers; there is only the transfer of internal thermal energy through fluids of different temperatures and between a liquid and a solid surface in thermal contact [7]. Heat exchangers are solely applicable for heating and cooling. The exchanger is the lifeblood of all devices. Therefore, it is essential to increase their effectiveness and quality [8,9]. Heat exchangers are crucial components in the majority of thermal applications in our daily lives [10], such as those for preserving food, cooling, and heating [11] and many other items that have an impact on our daily lives [12] such as in-home refrigeration devices [13] since these applications either need to absorb or reject

heat from a system, or do both [14,15]. As a result of all of this, engineers are searching for the most efficient means to boost heat transfer rates and maximize the effectiveness of these exchangers, as shown in Figure 1.

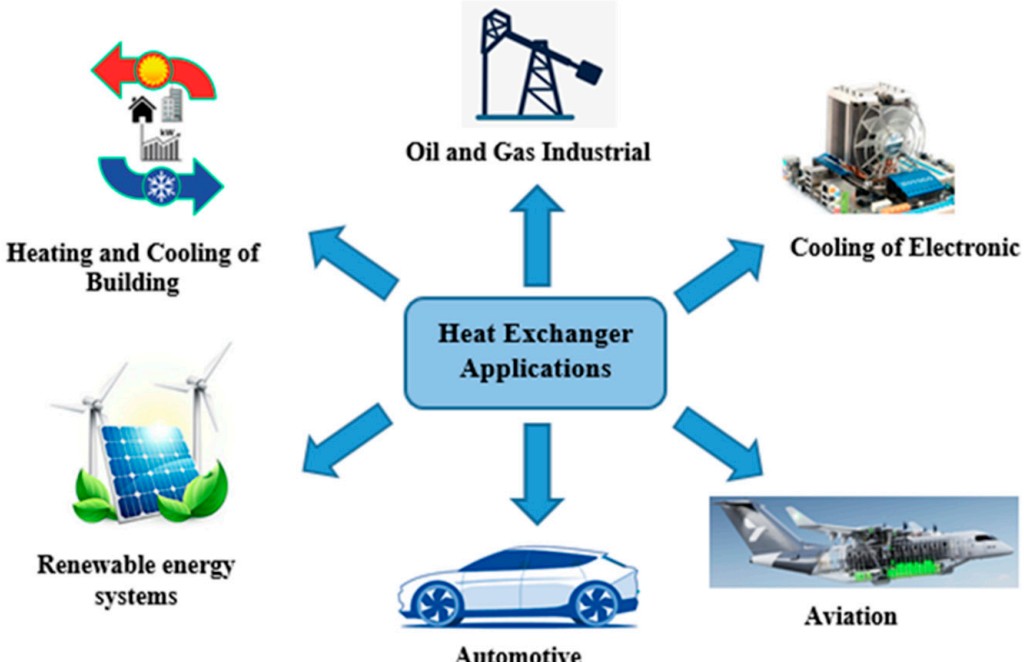

**Figure 1.** A few applications for heat exchangers.

The enhancement of heat transfer in all industrial applications is becoming increasingly important. Heat transfer enhancement techniques may be classified into three main classes, i.e., active, passive, and compound techniques. In active techniques, external power is used for heat transfer enhancement [15,16]. It seems an easy technique in several applications; however, it is quite complex from a design point of view. That is why it is of limited use due to external power requirements. In contrast, there is no involvement of external power supply in passive techniques of heat transfer enhancement. Passive techniques utilize the energy within the system, which leads to increased fluid pressure drop [16].

A number of techniques exist, including passive methods; these are used for monitoring process system thresholds and comparing them to values that have been determined in the past [17]. The following are included in this group: surface treatment, rough surfaces, increasing surface areas, swirl flow devices, twisted pipes, flow enhancement devices, additives for gases, and additives for liquids [18,19]. Active techniques work with a system's energy by introducing a few small adjustments or disturbances and evaluating how the system responds. Introduced disturbances may include vibrations of surfaces, a static electric field, the process of pulling or suctioning, injections, or extrusions [20], creating fins or jet collisions of liquids, and cooling gases in any bodies to be cooled [21]. Active and passive approaches may be combined to provide a third strategy for maximizing the heat exchanger's performance [22], as shown in Figure 2.

This review examines the latest research on heat transfer in heat exchangers, as well as its impact on performance and efficiency. It also discusses the advantages and disadvantages of these advances in energy transfer. With a focus on the correlative effects of these developments, this paper provides an updated overview of research on heat transfer in exchangers and recent breakthroughs in the field.

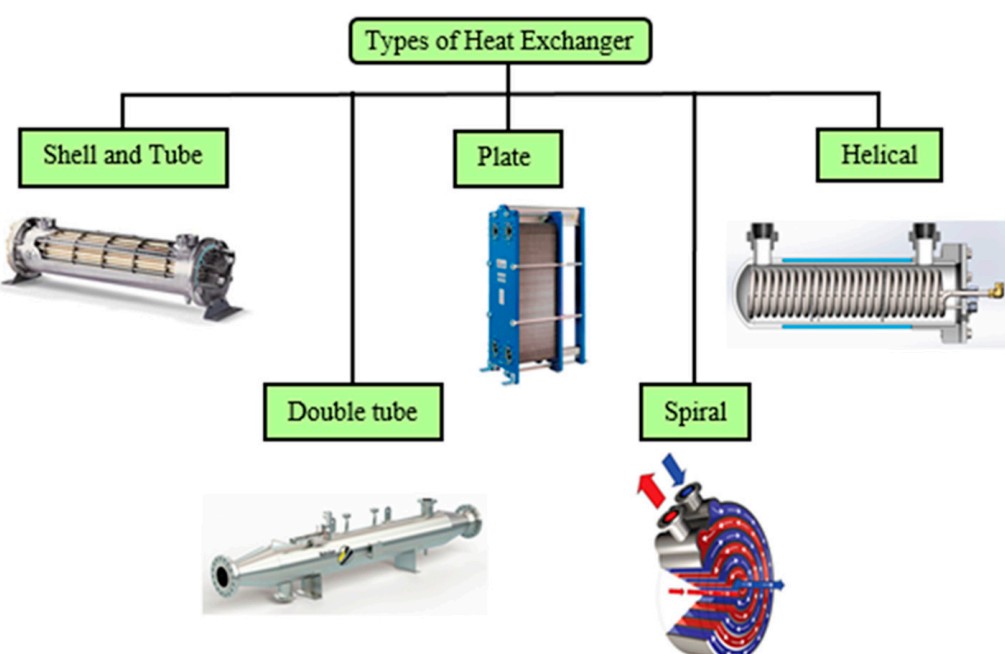

**Figure 2.** Frequently used heat exchangers in a variety of industries.

## 2. Heat Exchanger Enhancement and Classification

The most common method is the addition of twisted tapes and discs inside the exchanger tube to alter the shape of the flow and to increase flow disturbances to break the boundaries of the liquid layers and improve mixing between those layers [23]. Another method is to increase thermal conductivity by adding nanomaterials with high thermal conductivity. Because of their nanoscale, they can have the largest surface area possible, which boosts the conduction efficiency of these nanoparticles and liquids for these exchangers [24]. The highest thermal conductivity between the liquid layers can also be obtained by generating vortices and spiral flows, which intensifies the turbulence of the liquid and breaks the layers apart and creates vortices and spiral flows, which raises the intensity of the liquid's turbulence and increases fluid layer mixing and contact surface area between the fluid and the nanoparticles, which in turn increases the conduction efficiency of these nanoparticles and liquids for these exchangers to obtain the best thermal conductivity between the liquid layers [25,26]. Any pressure losses between the ends of the heat exchanger result in higher operating costs for these exchangers. The employment of these aids is intended to boost the efficiency of the exchangers, and when selecting nanomaterials, there should be conditions, such as there being no interactions between these nanomaterials and the base fluid, the cost and lifespan of the nanomaterials to achieve effective conductivity of heat exchangers, and the amount of their negative impact on the walls of the exchanger tubes [27]. All of this is an additional operating cost for heat exchangers. Naturally, the profit from the efficiency of the exchanger is an operational profit above all else, and the total operating profit is greater than any operating losses. The shape of the internal discs is altered to achieve the highest degree of crushing for the layer boundaries of the liquid and to create vortices that aid in better mixing of the liquid to achieve a higher conductivity between the layers of the liquid [28].

The researchers went further and changed the shapes of the twisted tapes in terms of their torsion ratio (such as the degree of torsion, the tape width ratio with the step per 360 degrees of torsion) [29,30], which in turn is reflected in increased heat transfer between the liquid and the tube with the adoption nanomaterials with high thermal conductivity [31]. These materials, as well as a nanomaterial, are introduced into the base liquid at different concentrations. One or more nanomaterials can be introduced to raise the

thermal conductivity of the fluid between the fluid layers and with the wall tube according to the concentration and type of nanomaterial [32].

The efficacy of heat exchangers in any system should, therefore, be taken into consideration while working to increase system efficiency, as it may not be advantageous to increase the efficiency of other systems if the exchangers do not have high efficiency. The researchers focused their research on improving heat exchanger efficiency as a result [33]. The majority of the methods developed in this discipline for enhancing heat transfer are one of two kinds [34]. Generally, after a survey of the historical development of heat exchangers, their classification can depend on several features.

1. Classification according to process function [30,32], as shown in Figure 3.

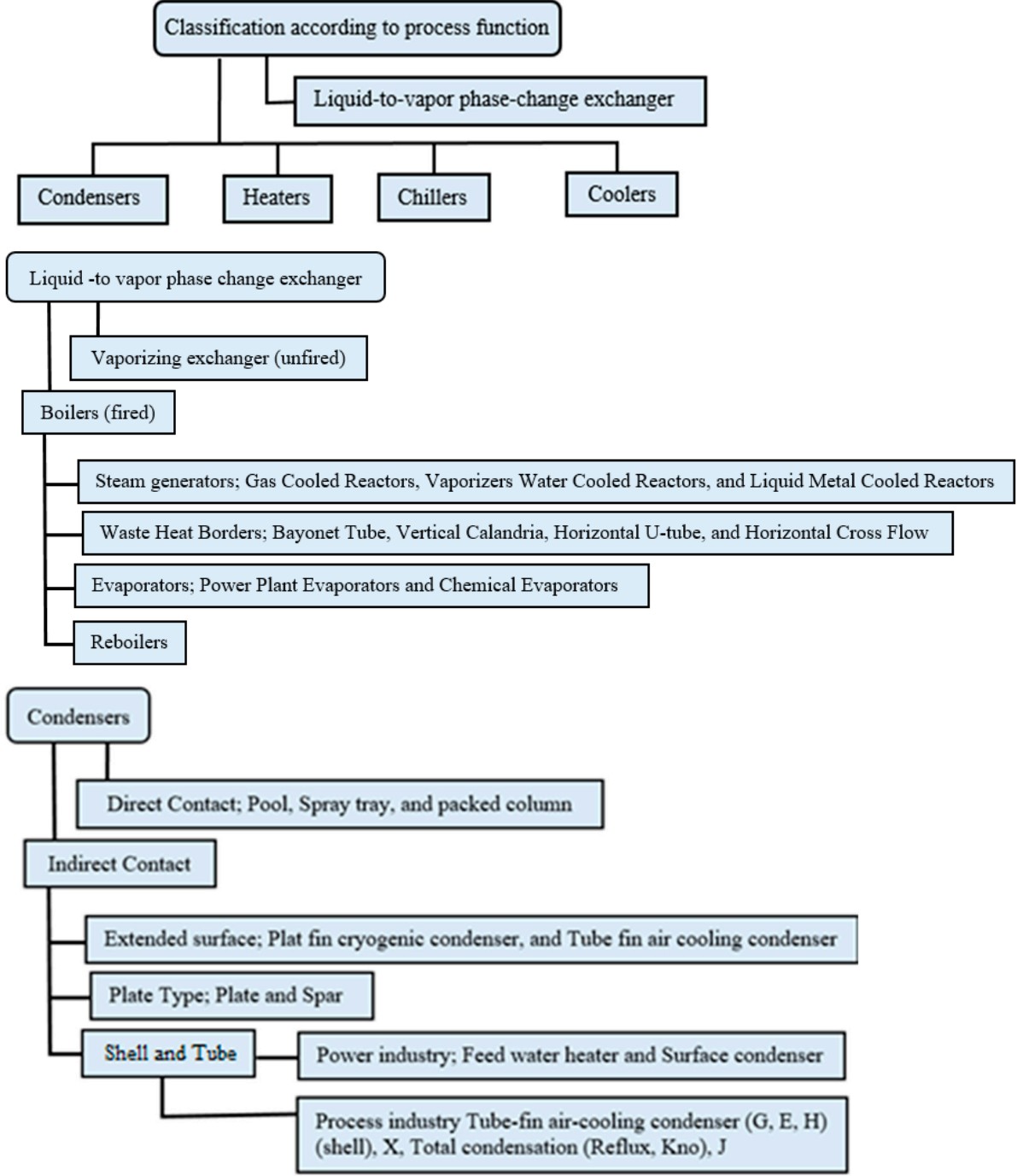

**Figure 3.** Flow charts of heat exchangers classification according to process function.

2.  Classification according to a type of construction [34,35], as shown in Figure 4.

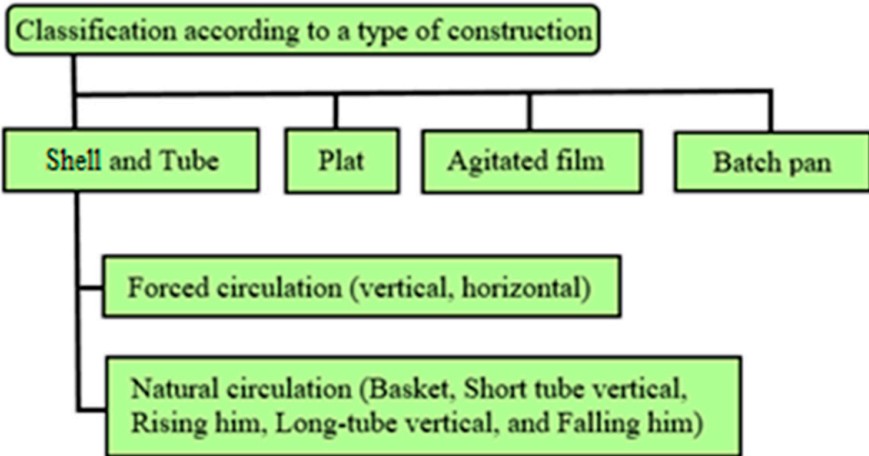

**Figure 4.** Flow chart of heat exchangers classification according to a type of construction.

3.  Classification according to how energy is supplied [36], as shown in Figure 5.

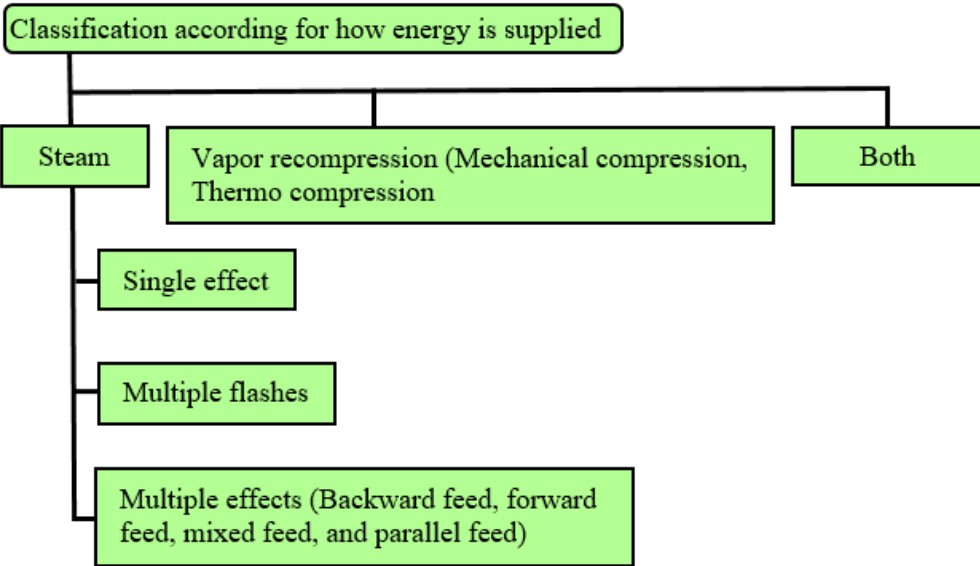

**Figure 5.** Flow chart of heat exchangers classification according to how energy is supplied.

4.  Classification of Reboilers [36], as shown in Figure 6.

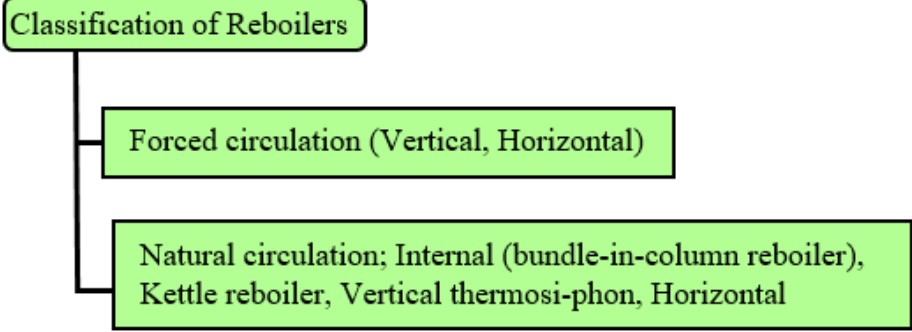

**Figure 6.** Flow chart of Reboilers classification.

## 3. Energy-Efficient Heat Exchangers

Heat exchangers have drawn researchers' attention to focus on improving the functioning of heat exchangers because of their widespread use and significant importance in all the systems of which they are a part. One of the most significant techniques applied and carried out in the studies is that of rings and twisted tapes.

### 3.1. Rings and Twisted Tapes

The general definition of boosting heat transfer is the study to improve heat transfer efficiency, an element of which is to lower the cost of heat exchanger equipment and accept a pressure loss or drop in exchange for an increase in heat transfer [37]. Discs and twisted tapes serve as inserts and can be used to update and develop heat exchangers in tubes to improve heat transfer; this improvement in the efficiency of a heat exchanger may be utilized to achieve the same efficiency with smaller-volume heat exchangers [38].

To increase the thermal performance of a heat exchanger and decrease system cost and size, substantial studies have been conducted on improving mechanisms of heat transfer. The generally used improvement technique in many engineering applications for heating and cooling systems is swirl/vortex flow, which has two categories: continuous swirl and decaying swirl. Swirls form at the entrance of a duct through generators, baffles, and winglet vortices [39]. With coiled wire inserts and helical/twisted tape, the entire length of a duct has a persistent swirling motion. To ease the mixing of the fluid layers, the thermal boundary layers of fluid inside the tube are typically ruptured using modified surfaces. Twisted tapes are utilized in areas near the tube wall where they operate as vortex generators and turbulence catalysts, extending the time that fluid flows remain in the tube and enhancing turbulence flow. Twisted tape is a popular way to improve heat transfer from the side of a tube due to its simple maintenance and inexpensive installation [40].

Experimental Studies on Ring and Twisted Tapes

Different authors have used a variety of twisted tape shapes to investigate laminar and turbulent flow in air and water. The literature review of numerous authors who have worked in this field of heat transfer enhancement is listed and reviewed. There are many different types of tape, such as twisted tapes with different geometries depending on the torsion ratio of the tape, which is a ratio between the twist of the pitch (H) and the thickness and width of the tape; the shape of the tape and whether it has holes; the shapes, measurements, and locations of those holes along the tape; the direction of the tape's twisting in each area; and other factors such as the metal type of the tape, the quantity of tape inside the tube, the existence of rings inside, the form of those rings, the presence of holes, the dimensions of the disks and holes, and spacing between disks. Akbari et al. [41] demonstrated, numerically studied, and determined the heat transmission and friction in a circular tube fitted with perforated twisted tape inserts having width ratios (W/D) in the range of 0.5–0.9, space ratios (C/D) of 0.30, 0.40, and 0.50, and twisted ratios (P/W) of 3, 3.5, and 4. At a fixed wall flux temperature, a water/$Al_2O_3$ nanofluid with volume percentage u = 0–4.0% of solid nanoparticles was utilized as a working fluid with Reynolds numbers between 500 to 25,000. According to the findings, using solid nanoparticles in turbulent flows at larger volume fractions and higher Reynolds numbers than in laminar flows enhances heat transmission. The improvement in heat transmission is significantly impacted by the presence of solid nanoparticles with lower twisted ratios (P/W). As the twisted tape grows wider and the nanoparticle concentration rises, heat transfer in the laminar flow is enhanced.

Kumar et al. [42] examined the impact of a passive technique, particularly in the lower Reynolds number range from 6500 to 26,500. The effect on heat transfer of perforated circular disk turbulators, friction factor, and heat exchanger thermal performance was investigated experimentally. The fixed diameter ratio of 0.8, pitch ratios of 1, 2, and 3, perforation indices of 0, 8, 16, and 24%, a fixed twist ratio of 2, and a fixed width ratio of 0.4 were some of the different geometrical characteristics employed for the experiment.

According to test findings, the heat transfer efficiency and thermal performance factor were significantly higher than those of smooth tube heat exchangers by a factor of 1.18 to 1.64, respectively. Kamel et al. [43] investigated the efficacy of a double-pipe heat exchanger with whole-length tight-fit twisted tape. To improve heat transfer, different twist ratios of y = 1.5, 2.0, 2.5, and 3.0 were selected. For the air side (inner tube), Reynolds numbers ranging from 25,000 to 92,000 and the flow rates for heavy fuel oil for the shell side of 0.1 kg/s were used. SOLIDWORKS PREMIUM 2016 software simulations were used to produce the ideal design. Schedule 10 m and 1.5 m long outer and inner stainless steel pipes have nominal sizes of 4 and 6 inches. The findings indicated that when compared to a plain tube, the greatest improvements in heat transfer rate, Nusselt number, and coefficient of convection heat transfer were approximately 26.6, 106.1, and 91.7%. Singh et al. [44] experimentally and computationally explored the effect of serrated circular rings with rectangular winglets (SCRWRW) on the thermal performance of a circular-tube heat exchanger. Diameter ratios (DR) of 0.8 and 0.85, pitch ratios (PR) of 3 and 4, attack angles of 15°, 30°, and 45°, and the blockage ratio of 0.05 were the geometrical parameters taken into account. The test section was a circular tube with a diameter of 68.1 mm and a length of 1.5 m subjected to a uniform heat flux of 1000 $W/m^2$. The operating fluid with a Reynolds number between 6000 and 24,000 was air. The findings revealed that SCRWRW increases the heat transfer rate 5.53 times faster than a smooth tube, with the maximum thermal performance being DR = 0.85, PR = 3, and an attack angle of 15° when thermal performance was at its highest. Utilizing ANSYS Fluent, the computational analysis also showed how SCRWRW affected the temperature distribution of the fluid.

Khoshvaght–Aliabadi et al. [45] examined the results of utilizing a VG (vortex generator) insert with various delta winglet configurations. Aluminum sheets measuring 350 mm × 14.5 mm and 0.6 mm thickness were used to create 14 VG inserts with a longitudinal and forward arrangement of delta winglets. Comparisons were made between the pressure drop and heat transfer data obtained while using the VG inserts inside the tube and those obtained from the plain tube. It was discovered that the Notter Rouse equation out predicts the Gnielinski equation for the current experimental Nusselt number at the transitional flow through the plain tube. Additionally, the experimental findings showed that using VG inserts inside the tube produced a maximum pressure drop and heat transfer coefficient compared to using the plain tube and that both parameters increased as the number of delta winglets increased. The optimal arrangement of the delta winglets on the VG insert, which exhibited the maximum heat transfer coefficient and the highest values of the examined PEC, was used to calculate the trade-off between the enhanced friction (performance evaluation criterion) and heat transfer. The VG insert's maximum PEC of 1.41 was discovered at Re = 148,715. Zhu and Chen [46] conducted a numerical analysis of the single-phase improved convective heat transfer effect using inserts made of (single, double, and triple) twisted tapes in heat exchange tubes. The airflow characteristics inside the tube during the heating process were examined using the wall surface enhancement equations and the second-order discrete (FVM) approach. When the outcome was compared with the traditional empirical equation, the error was found to be within +15%. According to the findings of the study, three distinct varieties of twist tape insert (TTI) can increase by 1.8 to 4.5 times a fluid's capacity to transmit heat. The three TTI types also displayed a variety of properties that affected how fluids flowed with more or less resistance. With a single TTI, the resistance can rise by 6.0 to 8.7 times, twice as much with a double TTI, and three times as much with a triplex TTI. Singh et al. [47] used the FLUENT module of ANSYS 16.0 for their computational investigation. A 1.5 m long and 68.1 mm internal diameter tubular heat exchanger provides a continuous heat flux of 1000 $W/m^2$. The working gas, air, has a Reynolds number (Re) ranging from 3000 to 21,000. For CCR rings, the Nusselt number and friction factor were computed after reaching steady-state conditions. Additionally, these rings were perforated with perforation indices of 8, 12, and 16%, with the thermo-hydraulic properties being researched. According to computational findings, perforating

the rings significantly affected the friction factor and heat transfer. Nu and TPF increased when the perforation index increased. For PCCR 16%, the highest improvements in Nu and TPF (as opposed to those of the smooth tube) were 7.4 and 1.56 times, respectively. Kharkwal and Singh [48] numerically and experimentally investigated how a circular ring with twisted tape might improve heat transmission. The work included calculating the friction factor, Nusselt number, thermal performance factor, diameter ratios 0.8 and 0.85, and pitch ratio variations for twisted tape 2 and 3 with serrated circular rings. The working fluid with a Reynolds number of 6000–24,000 was air. The experiment was carried out by continuously applying 1000 W/m$^2$ of wall heat flux to the system and measuring the outcomes at a steady state. The smooth tube case's computational and experimental results were compared with the typical Dittus Boelter and Blasius correlations. Compared to a smooth tube heat exchanger, there was a 5.16 times increase in heat transfer and a 3.05 times improvement in thermal performance, according to experimental and numerical research. In Promvonge [49], the tape was twisted with the main fluid, air, at a speed of Re = 3000 to 18,000 in a circular tube with a coil inserted. The results showed that when coils and twisted tapes are used together, heat transmission is double what it would be with only wire coils and twisted tape. Under the same circumstances, twisted tape with wire coils and the same twist with coil pitch ratios, heat transfers more quickly than with a bigger coil pitch ratio and twist. Promvonge et al. [50] worked on a tube with helical ribs with water as a working fluid and inserted twisted twin tapes pitched to tube diameter Y = 2.17–9.39. The results from both the smooth and ribbed tubes functioning alone were compared with those from the ribbed tube with twin twisted tape inserts. According to the experimental findings, the co-swirling inserted tube outperformed the ribbed–smooth tubes alone under similar working conditions. When Reynolds numbers are lower, the co-swirl tube at Y8 produces optimal thermal performance. Also known as co-swirl, the inserted ribbed tube is positioned with the twisted tape's helical swirl and the tube's helical rib motion moving in the same direction. Zaboli et al. [51], by numerically working on different lobe-shaped cross-sections for corrugated coiled tubes with a twisted spiral tape and speed equal Re = 35,000, found the thermal efficiency of three, four, and five lobe models with clear in the middle, twisted tape exceeds that of the three, four, and five lobe models with basic spiral twisted tape by 16, 18, 64, and 19.16%, respectively. Mahdi and Hussein [52], in experimental work and speed water Re = 400–2400 and double pipe with insert horizontal wing and cut twisted tape have a pitch to tube diameter = 2, 4.4, 6 find:

1.  Due to the effect of enhanced turbulence, performance is enhanced with horizontal wing-cut twisted tape, which improves the fluid mixing close to the test tube wall.
2.  PTT (Tapes), V-TT (V cut-TwistedTapes), and HW-TT (Horizontal Wing cut-Twisted Tape) have maximum thermal enhancement factors of 3.903, 4.269, and 4.488, respectively.

In Poongavanam et al. [53], the conical ring has a twisted tape with a pitch-to-tube diameter = 3.75, 7.5 inserted in a circular tube and air speed Re = 6000–26,000 and constant heat flux; an experimental work discovered for employing the twisted tape and conical ring of Y = 3.75. The efficiency of enhancement tends to decline for all employed devices as the Reynolds number increases and becomes virtually uniform above 16,000.

### *3.2. Nanofluids*

The approach for introducing nanomaterials to the fluids of heat exchangers that comprise rings and tapes twisted in various forms with varying quantities of nanomaterials for one or more types of nanomaterials was attempted to be summarized in this study. The major outcome that led to the suggested strategy is highlighted [53,54]. Due to the significant and abrupt rises in fuel prices over the past few decades, it has become necessary to increase the heat exchanger's conductivity and rate of heat transfer fluids to achieve high efficiency and small volume for those heat exchangers. The tendency was to suspend nanomaterials with high conductivity that did not surpass 100 nanometers in size, as

shown in Figure 7. Researchers worked tirelessly to increase the conductivity of the basic fluids utilized in heat exchangers and not subject the exchanger tube walls to any chemical interaction with the basic liquid substance by taking into consideration friction and the exchanger's pressure drops at both ends, as well as taking into consideration the pumping power of liquids. Note that the studies on nanofluids began in 1992 at the Argonne National Laboratory [55].

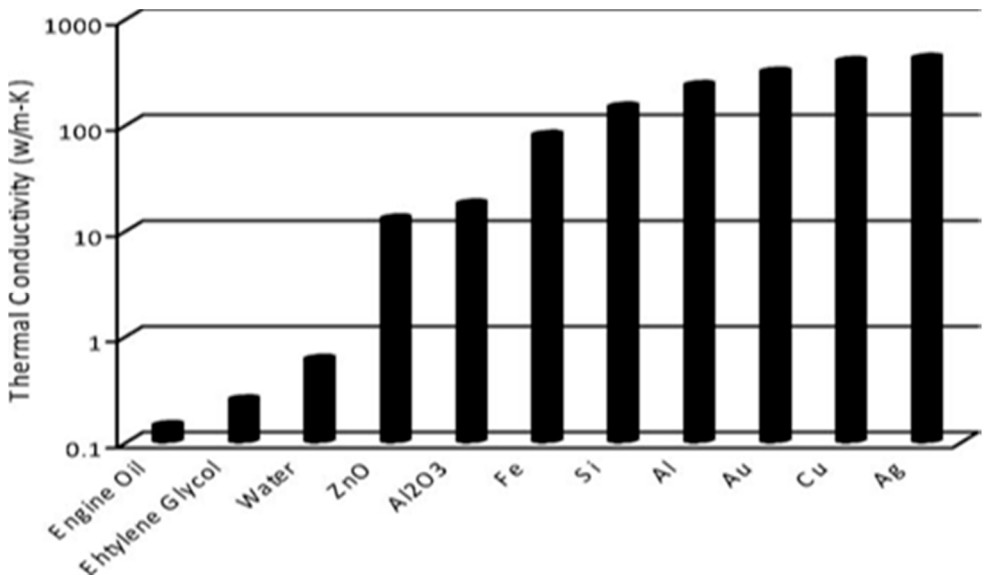

**Figure 7.** Thermal conductivity comparison of major liquids and solids [54].

The use of oil as a basic fluid for exchanges has increased due to the development of nanofluids. However, after adding nanomaterials to the oil and measuring the conductivity of the new nanofluid, it was found that the base liquids' conductivity was lower than that of the nanofluids [54,56]. The presence of metal particles and nanoparticle oxides results in nanofluids having a very high thermal conductivity [57]. The degree to which the liquid's thermal conductivity is improved depends on the nanoparticles' size and shape, as seen in Figure 8.

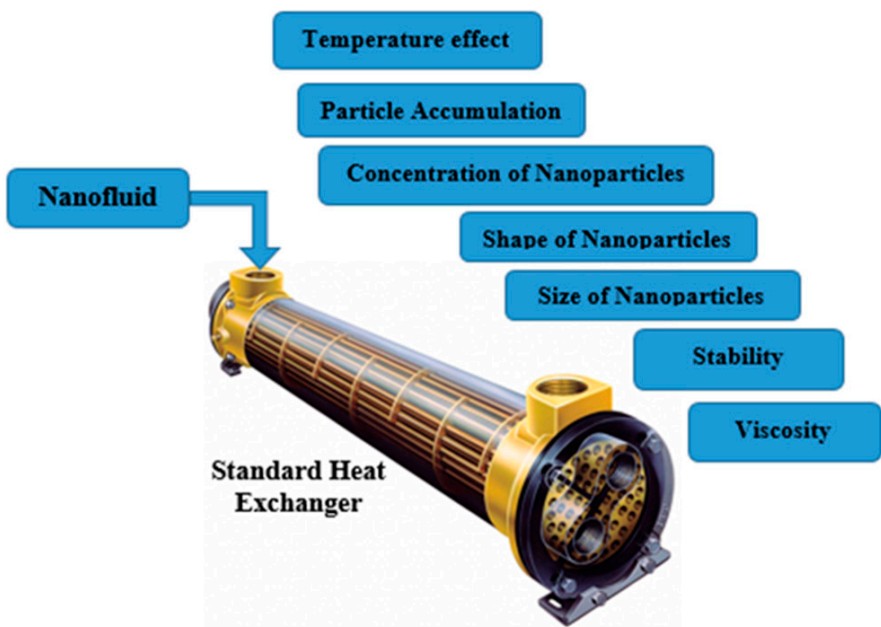

**Figure 8.** Physical factors influence how heat is transferred by nanofluids.

Experimental Studies on Used Nanofluids

According to research on ethylene, glycol, water, and saline liquids with concentrations of nanomaterials, including copper, aluminum, and silver, the thermal conductivity of nanofluids is linearly affected by nanomaterials with increasing concentration [58]. We increase the surface contact areas for these materials by utilizing nanoparticles with excellent conductivity due to the nanomaterials' extremely small sizes. Depending on the nanomaterial's concentration and thermal conductivity in the base liquid, the new fluid, known as a nanofluid, will have the maximum conductivity possible due to the high conductivity of the nanomaterials contained therein [59]. To statistically calculate the physical and thermal parameters of nanofluids, many researchers have discovered mathematical correlations. These equations are used to determine the physical characteristics of a nanofluid based on the physical characteristics of the nanomaterials and the base fluid employed, as well as their concentration and size within the base fluid. When the outcomes of these calculations were compared to the findings from the laboratory, there were relatively few variations, which ranged from 1.5 to 8% [60,61]. The specifications of the nanofluid can be determined using those numerical computations if these discrepancies are ignored [62]. In a numerical model, Azmi et al. [63] created twisted tape to insert into a tube to create a turbulent flow of nanofluids. The model was predicated on the idea that the Prandtl index and coefficient in momentum and heat can be used to apply the van Driest eddy diffusivity equation. For a wide range of Reynolds numbers Re, the outcomes of the results of the numerical analysis were matched to those found from experiments using $SiO_2$/water nanofluid. With the help of experimental data for twisted tapes, a generalized equation for the estimation of the Nusselt number and nanofluid friction factor was proposed. The numerical values as a function of the Reynolds number, concentration, and twist ratio were used to determine the Prandtl index and coefficient in the equation of the eddy diffusivity of heat and momentum. When compared to the flow of water inside the tube, the coefficient of heat transfer improved by 94.1%, and the friction factor increased by 160% at Re = 14–19,046 when the twist ratio was five. With $Al_2O_3$ and $Fe_3O_4$ nanofluids, there was a close interaction with the little experimental data from other studies, demonstrating the viability of the numerical model for use with twisted tape inserts [51,56,59]. Azmi et al. [64] determined by calculating the friction factor and heat transfer coefficients of $TiO_2$/water nanofluid at a temperature of 30 °C, up to a volume concentration of 3.0%. The investigations were conducted using tapes with various twist ratios and flow in tubes with Reynolds numbers varying from 8000 to 30,000. At a concentration of 1.0%, a significant increase in heat transfer coefficients of 23.2% was seen for a tube discharge. With the utilization of twisted tapes, the water and nanofluid heat transfer coefficient is enhanced with a decreasing twist ratio [65,66]. At Re = 23,558 and a twist ratio of 5 with 1.0% concentration, the friction factor and heat transfer coefficient were, respectively, 1.5 times and 81.1% greater than values with water flowing in a tube [63]. Heat transfer coefficients for flow in a tube and with tape inserts were reduced to levels lower than water with an enhancement in the nanofluid concentration to 3.0%. If pressure decreases and an increase in the heat transfer coefficient are considered, a thermal system with a tape insert with a twist-ratio of 1.5 and 1.0% $TiO_2$ concentration would offer the best benefit ratio [67] by inserting baffles into the twin-pipe heat exchanger's annulus.

Singh et al. [68] found the properties of CuO water-based nanofluids. Copper oxide and still water, water-based nanofluids were used in the experiment at volume concentrations of 0.1 and 0.2. The effect of temperature on the heat transfer coefficient and the Nusselt number was investigated in this experiment. It was found that copper oxide nanofluids with baffles would exhibit greater enhancement of heat transfer than pure water. Without adding baffles and nanofluids, the Nusselt number increased by 8%, while a 10–12% enhancement was seen when baffles and nanofluids were used. The rate of heat transfer was found to be enhanced by 22–25% at 0.1% volume concentration and by 25–30% at 0.2% volume concentration. By employing a nanofluid of $Fe_3O_4$,

Krishna et al. [69], using $Fe_3O_4$ water-based nanofluid, conducted an experimental examination of twisted tape inserts with different cut-radii in hairpin heat exchangers. Data from experiments were produced at mass flow rates between 0.05 kg/s to 0.25 kg/s and at Reynolds numbers within 3,000 to 25,000, wherein the hot fluid in the annulus was kept moving at a constant pace of 0.1 kg/s. For several particle volume concentrations between 0.011% and 0.031%, performance analysis required calculating the heat transfer coefficient and related friction factors. With a cut radius of r = 6 and insert of H/D = 3, the Nusselt number for the entire pipe for 0.03% nanofluid concentrations increased by 32.01% above water. In comparison with water, the friction factor of the entire pipe increased by 1.21 times at nanofluid concentrations of 0.03%, r = 6, with twisted tape inserts, and H/D = 3. Following a rise in the nanofluid's volume concentration, the findings of the investigation demonstrated that the heat transfer coefficient and friction factor, a performance characteristic of the heat exchanger, is improved [67,68]. Qi et al. [70] investigated the thermo of hydraulic behavior of $TiO_2$ and $H_2O$ nanofluids in triangle-shaped tubes when taking the impact of twisted tapes into account. To estimate this system, energy and thermal efficiency theories were applied. An experimental method was conducted on the effects of different triangular tube structures (tube 2) isosceles 45° triangle tube, (tube 1) isosceles right triangle tube, Reynolds numbers vary from 800 to 9000, mass fractions of nanoparticles differ from 0.1 to 0.3 and 0.5%, and twisted tape, Nusselt number, resistance coefficient, Nusselt number ratio, pressure drop, and resistance of coefficient ratio. The research was carried out on the performance of thermo-hydraulic $TiO_2$ and $H_2O$ nanofluids in triangle tubes considering the twisted tape's effect. This system's estimation was based on the theories of thermal and energy efficiency. Using silicon carbide nanofluids as a coolant, Kazem et al. [71] investigated the efficacy of a photovoltaic/thermal (PV/T) system. Testing was conducted in Oman with nanoparticles weighing 0.5%. The thermal conductivity, density, and viscosity of the liquid increased by 6.64%, 13%, and 12%, respectively, at 25 °C. Furthermore, the effect of increasing density and viscosity (by adding nanoparticles to the base fluid) was investigated. According to the results, turbulent flow decreased the pumping force, whereas laminar flow increased it. Singh and Kumar [72] researched the evaluations of several kinds of TT with the use of double-pipe heat exchangers with different nanofluids. The discussion of the friction factor (f) and the heat transfer rate was extensive. When compared to turbulent flows, twisted tape inserts were shown to be more efficient in laminar flows. On the other hand, heat transfer rates in turbulent flows were higher than those in laminar flows when solid nanoparticles with a high value were used [69,70]. Prasad et al. [40], through an applied study, studied the behavior of nano-liquid water with aluminum oxide at a concentration of 1.3 and speeds between the Reynolds number = 3000–30,000. For a tube, double pipes in a U-tube have trapezoidal cuts in twisted tape inserts. The results of the experiment show that when the volume concentration of the nanoparticle increases, the heat exchanger's heat transfer coefficient and friction factor also rise. He et al. [73] investigated the hypothesis of the behavior of nanofluid water containing copper oxide at a concentration of 1–4 and speeds ranging from 3,000 to 36,000 Reynolds numbers for a circular tube with twisted tape. When compared to the single-phase model, the two-phase mixing model produced results that were more accurate in reality. From the observation of the attached Figures 9 and 10, the process of change in the friction factor and Nusselt number with fluid velocity (Reynolds number) is noted.

In Sadeghi et al. [74], aluminum oxide, silica oxide, and water nanoparticles were combined at a Reynolds number = 100–10,000 within a circular tube for experimental and theoretical research. The tube had a helical tape insert with a 1.95 to 4.89 pitch-to-tube diameter ratio. The two adjacent figures, in sequence, show each Nusselt number and friction factor pressure drop while noting that $Al_2O_3$-containing nanofluids had the greatest Nusselt number. In every situation that was looked at consequently, when the Reynolds number grew and the twist ratio of the helical tape insert decreased, the Nusselt

number increased [75]. In Azmi et al. [64], a practical study was conducted on a mixture of nanomaterials of titanium oxide with silica oxide at a concentration of 0 to 3 with water at Reynolds number = 5,000–25,000 inside a circular tube. The tube contains a twisted tape insert with a pitch-to-tube diameter from 5 to 93. The Nusselt number is shown in Figure 11, noting that with the twist ratio, the augmentation of heat transport is inversely increased [63].

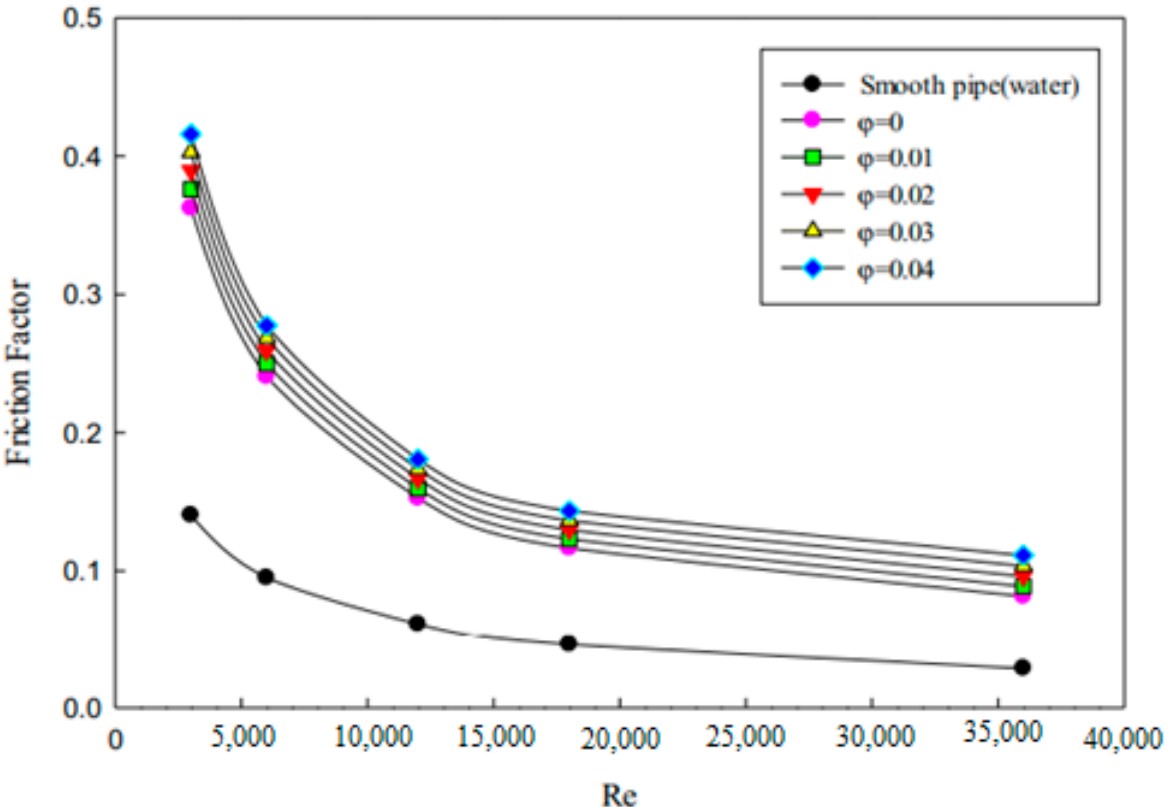

**Figure 9.** Friction factor versus Reynolds number [73].

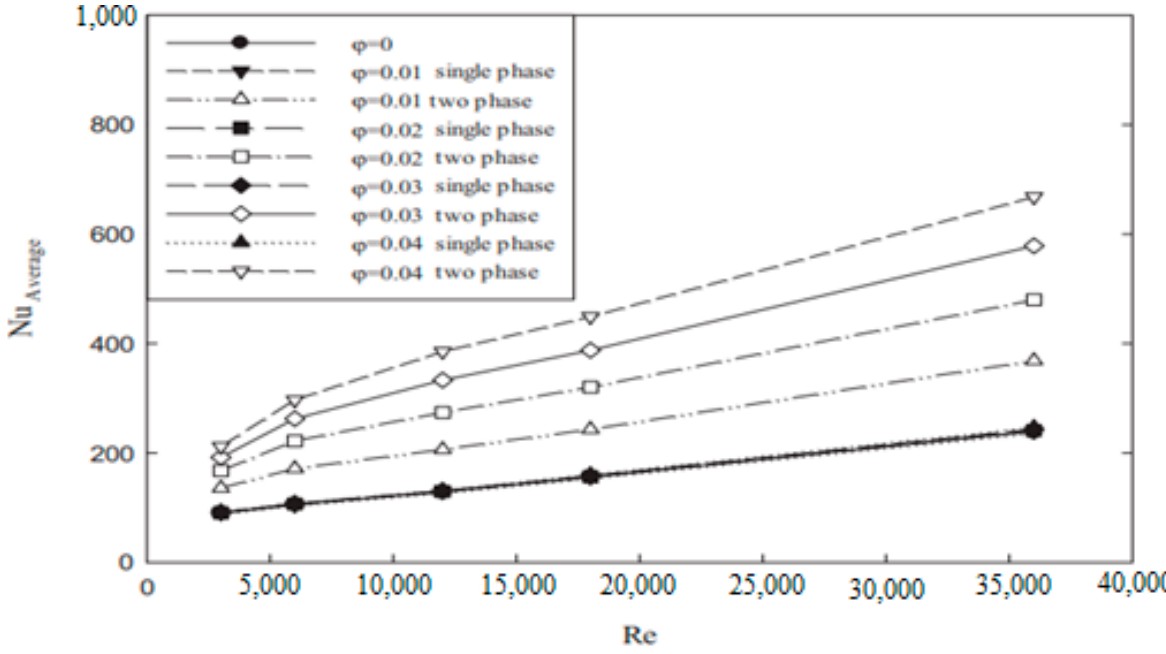

**Figure 10.** Average Nusselt number versus Reynolds number [73].

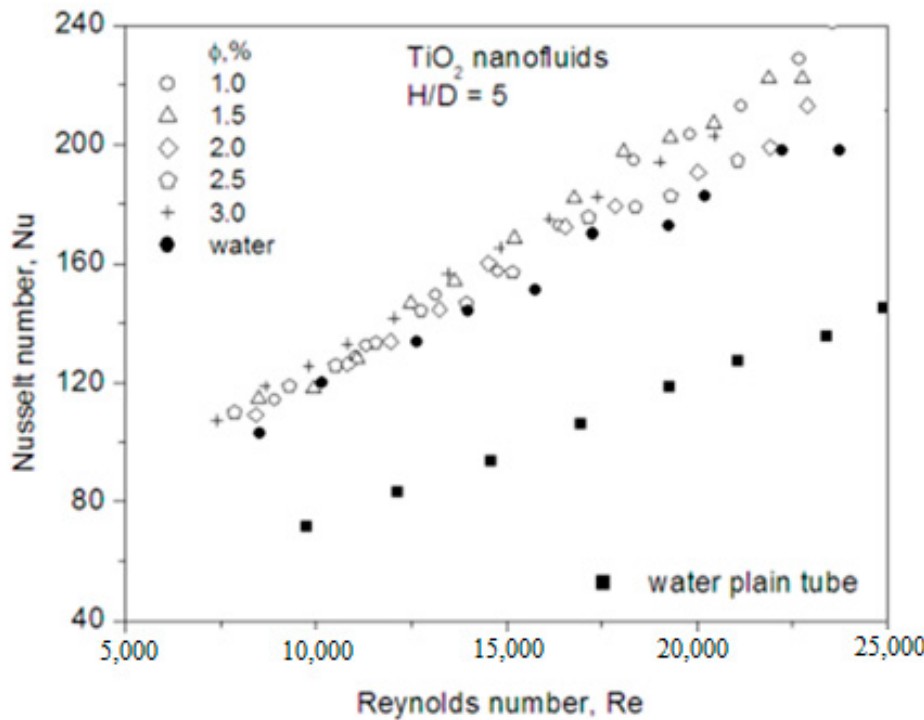

(**a**)

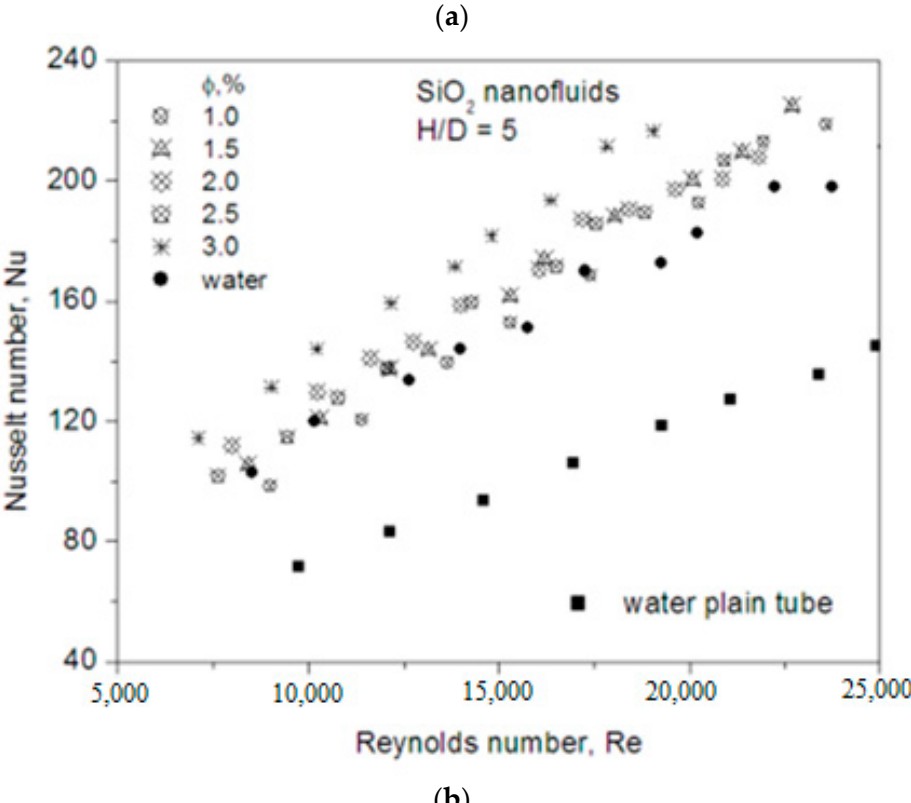

(**b**)

**Figure 11.** Average Nusselt number versus Reynolds number; (**a**) TiO$_2$ nanofluids, and (**b**) SiO$_2$ nanofluids [63].

## 4. Conclusions

The studies presented above all used empirical and numerical analyses to accelerate heat transfer in heat exchangers; however, no field of study was unified to break the

boundaries of liquid layers and improve the mixing process between liquid layers or to increase the efficiency of heat exchangers. It is normal for there to be a pressure drop between the ends of the heat exchanger tubes. To reduce this, differently shaped slots are placed in different places along the tapes and inside the discs, and some of these slots cause further improvements in raising the efficiency of the heat exchanger while reducing the pressure drop. Increasing the efficiency of the exchangers again, the process of converting the basic fluid of the exchanger into a nanoliquid using nanomaterials with high thermal conductivity and with variable concentrations and different types of nanomaterials to increase the thermal conductivity between the layers of the liquid all increased efficiency as a result.

As a result, much research is still being conducted to develop tape and disc forms that may be placed inside tubes for heat exchangers to maximize heat transmission with the least amount of pressure loss. Following up on earlier research and investigations led to the following discoveries.

1. Any increase in the efficiency of the heat exchanger is made possible by two opposing properties (a reduction in the pressure drop and an increase in the amount of heat being transferred).
2. Several new techniques are being developed by researchers for twisting tapes, including pinching or breaking twisted tapes and the use of nanofluids.
3. The majority of studies have looked at high Reynolds numbers that indicate an increase in energy consumption. Despite the use of techniques to increase the efficiency of the heat exchanger through helical flow and the amount of disturbance in the flow patterns of the liquid layers, particularly in the layers that come into contact with the exchanger tube wall, the increases destroyed the liquid layer borders and boosted the effectiveness of mixing between these layers.
4. The majority of research comparing the effectiveness of the exchanger with the presence of rings and tapes or tapes only inside the exchanger tube did not mention the difference in the volume of liquid inside the tube that will change for the few lengths of the exchanger that can be disregarded. However, if the exchanges are large, this difference will affect the outcomes.
5. It was not mentioned in many studies that the tapes should be fixed inside the tube to ensure the freezing of their position and their effect on the liquid inside the tube because, if they were left free, their location and effect would change with the disturbance of the fluid flow and possibly cause a few vibrations.

## 5. Future Research Proposals

1. Designing twisted and ring bands in forms and dimensions that maximize exchanger performance while minimizing pressure drop losses at both ends of the heat exchanger.
2. Investigating the same old designs, but altering the Reynolds numbers or the dimensions of the previous forms following the past research, to find the modifications that may increase the exchanger's effectiveness.
3. Examining the use of novel nanomaterials or examining the effects of more seasoned ones in various concentrations and for various base fluids, as long as safety considerations are taken into account while selecting the nanomaterials and base liquid.
4. In addition to this, the use of twisted tapes with nanofluids and other geometries is also found to be very active. However, more future work in this area will be very effective in improving this technology, and more realistic useful correlations will be obtained.
5. It is important to find a nanofluid that has high thermal conductivity in addition to appropriate density and viscosity so that the fluid can be circulated through the heat exchanger without losing energy.

**Funding:** This research received no external funding.

**Institutional Review Board Statement:** Not applicable.

**Informed Consent Statement:** Informed consent was obtained from all subjects involved in the study.

**Data Availability Statement:** Not applicable.

**Conflicts of Interest:** The authors declare no conflict of interest.

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
