# Peer review of "Impact of Nano Additives in Heat Exchangers with Twisted Tapes and Rings to Increase Efficiency: A Review"

_sustainability, doi:10.3390/su15107867_

Round 1
Reviewer 1 Report
The proposed paper is well-organized and worth to be published. However, some minor corrections should be made before publishing.
1. The quality of some figures should be improved, i.e. Fig. 3, Fig. 8
2. notation of numbers over 1,000: authors once use commas and once not; this should be standardised
3. The authors should also standardise the way units are written after numbers, such as 8% or 8% (with or without spaces)
The text, in its entirety, should be rechecked with editorial integrity.
Author Response
Respond to Reviewer 1 comments
The proposed paper is well-organized and worth to be published. However, some minor corrections should be made before publishing.
- The quality of some figures should be improved, i.e. Fig. 3, Fig. 8.
Authors: The reviewer comment is valid, and it has been considered.
- Notation of numbers over 1,000: authors once use commas and once not; this should be standardized.
Authors: Done. It has been considered.
- The authors should also standardize the way units are written after numbers, such as 8% or 8% (with or without spaces)
Authors: the reviewer comment is valid, it has been considered.
- The text, in its entirety, should be rechecked with editorial integrity.
Authors: the reviewer comment is good.

Reviewer 2 Report
General Comments: The manuscript deals with the “Impact of Nano Additives in Heat Exchangers with Twisted Tapes and Rings to Increase Efficiency: A Review”. The paper is straightforward and well written. It can be accepted if the authors address the following points.
COMMENTS: From the study of this work, I have come to the following conclusions:
My comments are as follows.
• Avoid lumping references together in the whole manuscript.
• Authors should check the typo, grammar and the formula of the manuscript.
• Too many references for a research paper. Some general and old references can be deleted. Use the following references for strengthening your manuscript:
o https://doi.org/10.1016/j.powtec.2019.01.081
o https://doi.org/10.1016/j.renene.2019.03.059
o https://doi.org/10.1016/j.tsep.2019.100403
• The main objective of the work must be written on the clearer and more concise way at the end of introduction section.
• It is better to provide space between numerals and units/ maintain uniformity in using SI units.
• All equations are in Math type format.
• Kindly provide the citation for all equations, which are not derived by the authors.
• Change the word shall and tube into Shell and tube in Figure 3 and 4.
• Line number 400 Change the “Nussault number” spelling.
• Give the clear diagram in Figure 11. And also mention Figure (a) and (b).
Major revision as briefly mentioned above is required for further consideration.
Author Response
Respond to Reviewer 2 comments
General Comments: The manuscript deals with the “Impact of Nano Additives in Heat Exchangers with Twisted Tapes and Rings to Increase Efficiency: A Review”. The paper is straightforward and well written. It can be accepted if the authors address the following points.
COMMENTS: From the study of this work, I have come to the following conclusions:
- Avoid lumping references together in the whole manuscript
Authors: the reviewer comment is valid, and it has been considered.
- Authors should check the typo, grammar and the formula of the manuscript
Authors: the reviewer comment is valid, and it has been considered.
- Too many references for a research paper. Some general and old references can be deleted. Use the following references for strengthening your manuscript
- https://doi.org/10.1016/j.powtec.2019.01.081
- https://doi.org/10.1016/j.renene.2019.03.059
- https://doi.org/10.1016/j.tsep.2019.100403
Authors: the reviewer comment is valid, and it has been considered. We added these references.
- The main objective of the work must be written on the clearer and more concise way at the end of introduction section.
Authors: Done.
- It is better to provide space between numerals and units/ maintain uniformity in using SI units.
Authors: the reviewer comment is valid, and it has been considered.
- All equations are in Math type format.
Authors: the reviewer comment is valid.
- Kindly provide the citation for all equations, which are not derived by the authors.
Authors: Done.
- Change the word shall and tube into Shell and tube in Figure 3 and 4.
Authors: Done. These words have been changed to figures.
- Line number 400 Change the “Nussault number” spelling.
Authors: Done.
- Give the clear diagram in Figure 11. And also mention Figure (a) and (b).
Authors: the reviewer comment is valid, and it has been considered.

Reviewer 3 Report
· Keywords could be improved. Repetitive words could be removed.
· The abstract is a standalone part of the paper. The abstract could be improved and revised. The problem statement, underlying research question, need for the review, objective of the review, significant outcomes of the review, and target beneficiaries can be added to increase the quality. This will help the readers to understand the scope of the paper.
· Figures 1 and 2 are not professional and has not have the quality for the review paper. Authors can reproduce Figure 2, instead of taking the figure from the publication.
· After Figure 1, it is mentioned that ‘Many techniques include the following…….’ This para needs modification. Passive and Active techniques must be mentioned first before explaining the techniques.
· The introduction, if possible, could be broken down into subsections. It is difficult to follow the topics discussed with continuous paragraphs in their current form.
· In my opinion, the classification mentioned in Figures 3-6 could be a separate section. Why is Figure 3 presented in a different colour?
· In the introduction, please discuss the previous reviews on the research topic and clarify how your review is different from the other published reviews. This would provide clarity on the novelty of the review.
· A diagram/workflow could be added in the Methodology section to show how the papers were selected for the review.
· Subsections 2.1, and 2.2 are too long with continuous paragraphs. The authors have just summarized the works done on the heat exchangers and have not provided their insights and scientific discussion is lacking.
· Point 2 in conclusion ‘No specific area of research has been mentioned’ is not clear.
· Section 4 is general and I recommend the authors have more technical discussion and suggestions for future research.
Author Response
Respond to Reviewer 3 comments
- Keywords could be improved. Repetitive words could be removed.
Authors: reviewer comment is valid, and it have been considered. We changed keywords.
Keywords: Heat exchanger; twisted tape; insert rings; Nusselt number; Nanofluid
- The abstract is a standalone part of the paper. The abstract could be improved and revised. The problem statement, underlying research question, need for the review, objective of the review, significant outcomes of the review, and target beneficiaries can be added to increase the quality. This will help the readers to understand the scope of the paper.
Authors: Done. The abstract have been changed.
- Figures 1 and 2 are not professional and has not have the quality for the review paper. Authors can reproduce Figure 2, instead of taking the figure from the publication.
Authors: the reviewer comment is valid, and it has been considered. We changed Figures 1 and 2.
- After Figure 1, it is mentioned that ‘Many techniques include the following…….’ This para needs modification. Passive and Active techniques must be mentioned first before explaining the techniques.
Authors: the reviewer comment is valid, and it has been considered. We added para;
“In the current time, the matter of heat transfer enhancement has more vital in all industrial applications. Especially, heat transfer enhancement techniques may be classified into three main classes i.e., active, passive and compound techniques. In active technique, external power is used for heat transfer enhancement [15, 16]. It seems an easy technique in several applications however it is quite complex from design point of view. That is why it is of limited use due to external power requirements. Apart from active techniques, there is no involvement of external power supply in passive techniques of heat transfer enhancement. Passive techniques utilize energy within the system which leads to increase fluid pressure drop [16]. Many techniques include the following…….”
- The introduction, if possible, could be broken down into subsections. It is difficult to follow the topics discussed with continuous paragraphs in their current form.
Authors: the reviewer comment is valid, and it has been considered. We divided introduction.
- In my opinion, the classification mentioned in Figures 3-6 could be a separate section. Why is Figure 3 presented in a different colour?
Authors: the reviewer comment is valid, and it has been considered. We changes colour.
- In the introduction, please discuss the previous reviews on the research topic and clarify how your review is different from the other published reviews. This would provide clarity on the novelty of the review.
Authors: the reviewer comment is valid, and it has been considered.
- A diagram/workflow could be added in the Methodology section to show how the papers were selected for the review.
Authors: the reviewer comment is valid, and it has been considered.
- Subsections 2.1, and 2.2 are too long with continuous paragraphs. The authors have just summarized the works done on the heat exchangers and have not provided their insights and scientific discussion is lacking.
Authors: Thank you on this comment.
- Point 2 in conclusion ‘No specific area of research has been mentioned’ is not clear.
Authors: the reviewer comment is valid, and it has been considered.
- Section 4 is general and I recommend the authors have more technical discussion and suggestions for future research.
Authors: The comment is really appreciated. This added value to the work.

Round 2
Reviewer 2 Report
The article can be accepted in the present form for publication.
Reviewer 3 Report
-
